# Biomass-Derived Porous Carbon Materials for Li-Ion Battery

**DOI:** 10.3390/nano12203710

**Published:** 2022-10-21

**Authors:** Meruyert Nazhipkyzy, Anar B. Maltay, Kydyr Askaruly, Dana D. Assylkhanova, Aigerim R. Seitkazinova, Zulkhair A. Mansurov

**Affiliations:** 1Department of Chemical Physics and Material Science, Al-Farabi Kazakh National University, Almaty 050040, Kazakhstan; 2Institute of Combustion Problems, Almaty 050012, Kazakhstan; 3Department of Materials Science, Nanotechnology and Engineering Physics, Satbayev University, Almaty 050000, Kazakhstan

**Keywords:** sawdust, electrospinning, fibers, anode, battery

## Abstract

Biomass-based carbon nanofibers (CNF) were synthesized using lignin extracted from sawdust and polyacrylonitrile (PAN) (30:70) with the help of the electrospinning method and subsequent stabilization at 220 °C and carbonization at 800, 900, and 1000 °C. The synthesized CNFs were studied by scanning electron microscopy, energy-dispersive X-ray analysis, Raman spectroscopy, and the Brunauer–Emmett–Teller method. The temperature effect shows that CNF carbonized at 800 °C has excellent stability at different current densities and high capacitance. CNF 800 in the first test cycle at a current density of 100 mA/g shows an initial capacity of 798 mAh/g and an initial coulomb efficiency of 69.5%. The CNF 900 and 1000 show an initial capacity of 668 mAh/g and 594 mAh/g, and an initial Coulomb efficiency of 52% and 51%. With a long cycle (for 500 cycles), all three samples at a current density of 500 mA/g show stable cycling in different capacities (CNF 800 in the region of 300–400 mAh/g, CNF 900 and 1000 in the region of 100–200 mAh/g).

## 1. Introduction

The diminishment of fossil energy and the necessity of protecting the environment requires the development of new devices that can store renewable energy [1]. Electrochemical energy storing devices (such as supercapacitors, lithium-ion batteries, etc.) have been studied a lot due to their fast charge-storing ability (i.e., low discharge time: 1–10 s for supercapacitors (SCs), 10–60 min for Li battery (LiB)) and enhanced cyclic stability (SCs > 30,000 h, battery > 500 h) [2].

The most crucial factor for electrochemical energy storage devices is the electrode material that shows their charge-storing ability. According to the research, the traditional powder electrode with an active material coating is high-priced and has a low utilization rate of the active material, a short service life, and other defects [3]. Consequently, new suitable inexpensive and high-efficiency electrode materials are necessary to be found to produce electrochemical energy-storing devices [4,5,6,7]. 

At present, batteries and SCs are two main energy-storing technologies. Li-ion batteries possessing a high energy density (~300 W h kg^−1^) made a substantial contribution to this field, and were awarded with the Nobel Prize in Chemistry 2019; but, there are some disadvantages. The working life of these batteries is limited (a few thousand cycles), they have a slow recharge rate, and low power density, and they should also be used with a great care [8,9].

The best material for electrochemical energy-storing devices is electrospun carbon nanofibers (ECNFs) [10,11,12]. ECNFs belong to a class of one-dimensional carbonaceous materials that are excellent electronic conductors. This quality allows using them as conductive additives in electrode materials for the LIBs. In addition, electrospun fibers can be obtained from various polymer precursors. The most important precursor in carbon fiber production is PAN, which requires only a laboratory unit to obtain carbon filaments [13]. The cost-effective and convenient electrospinning of organic polymers can also manufacture CNFs after carbonization [14,15,16]. Nanofibrous carbon-based electrodes are the main constituents in lightweight and environmentally friendly batteries. 

The overall performance-determining material for lithium batteries is the anode, which is one of their most vital components. Currently, widely spread materials used for anodes for commercial LIBs are graphite powders with a limited theoretical capacity (372 mA h g^−1^) and long diffusion pathways for lithium ions [17]. This may lead to low energy and low power densities, which does not correspond to the ever-growing demands for next-generation LIBs. The problem was solved by studying different nanostructured carbonaceous materials as anode materials for LIBs, such as carbon nanobeads [18], hollow carbon nanospheres [19], carbon nanotubes [20,21,22], carbon nanofibers (CNFs) [23,24,25], graphenes [26], and their composites [27,28,29].

In the mid-2000s, CNFs found their first use as anode materials for lithium batteries. Zou et al. [30] and Kim et al. [31] described the lithium storing ability of carbon nanofibers. Although earlier papers illustrated the potential application of carbon nanofibers in LIBs, their electrochemical characteristics were only 220 mAh/g after 30 cycles at 0.1 mA cm^−2^ and 450 mAh/g after 2 cycles at 30 mAg^−1^. Therefore, the upgrading of the electrochemical ability of anode materials based on CNF is of great relevance. 

Chunhui Chen et al. made a comparison of CNFs synthesized at 800 °C for one hour with activated (with KOH) CNFs as the anode for an LIB. ACNF manifested a reversible capacity of 512 mAh/g at the 100th cycle at a current density of 100 mA g^−1^. At a current density of 1000 mA g^−1^, the reversible capacity was 265 mAh/g [32]. Chan Kim et al. demonstrated that carbonization of CNTs at 2800 °C had a relatively developed texture, but a greatly reduced electrical conductivity (from 55.41 to 20.15 S cm^–1^). It was caused by the diminishment of the number of contact points between nanofibers because of a sharp jump volume [31]. The specific capacity of carbonized CNTs at 2800 °C at a current density of 100 mA g^−1^ was equal to 130 mAh/g. Lei Tao et al. described the effect of the carbonization of CNTs to utilize them as flexible anode materials for lithium-ion batteries. CNF carbonization at temperatures of 1500 and 2000 °C ended in a smaller specific surface area, a graphite-like layer, and a dramatic rise in conductivity. The specific capacity of carbonized CNT at1500 °C shows 182.5 mAh/g and CNT at1200 °C shows 107.7 mA/g [33]. 

In [34], the authors described the preparation of porous carbon/manganese oxide (C/MnOx) composite nanofibers from ultra-thin polyacrylonitrile (PAN)/Mn(CH_3_COO)_2_ (Mn(OAc)_2_) nanofibers with the help of a relatively simple and low-costing electrospinning method together with subsequent thermal processing, due to which they become the most suitable variants for anode materials to produce high-performance rechargeable lithium-ion batteries. This resulted in larger charge (597, 645, and 753 mAh/g) and discharge capacities (586, 645, and 714 mAh/g) of C/MnOx nanofiber anodes (15, 30, and 50 wt% Mn(OAc)_2_ in a precursor, which corresponded to coulombic efficiencies of 98.2%, 100%, and 94.8%. However, a pure CNF anode has a much lower reversible capacity (about 396 mAh/g), corresponding to a capacity retention of 73.0%.

Graphite is currently the most successfully commercialized anode material. However, its limited theoretical capacity and limited power density seem to be insufficient for the next generation LIBs [17]. To overcome these problems, new materials with fundamentally higher capacitance and higher power density are urgently needed. Recently, there has been a growing interest in the development of new carbon nanomaterials to replace graphite as anode materials for LIBs. These materials include carbon nanofibers obtained from wood waste and PAN polymer by electrospinning.

This paper focuses on the application of CNFs carbonized at a temperature of 800 °C, 900 °C, and 1000 °C as an anode in LIBs. The obtained fibers and electrodes were fully studied by electrochemical measurements, cyclic voltammetry (CV), impedance spectroscopy and cycling performance tests, SEM analysis, XRD, and Raman spectra. 

## 2. Experimental 

### 2.1. Materials and Chemicals

Sawdust was collected from Almaty (Kazakhstan). The chemicals used in this study were of analytical grade and included hydrochloric acid (HCl, 36.6%), formic acid (98–100%), potassium hydroxide (KOH, ≥85%, Sigma Aldrich, St. Louis, MI, USA), polyvinylidene fluoride (PVDF, EQ-Lib-PVDF, MTI Corporation, Richmond, CA, USA), conductive carbon black (EQ-Lib-SuperC45, MTI Corporation), 1-methyl-2-pyrrolidone (NMP, ≥99.0%, Sigma Aldrich), ethanol (99.5%, Sigma Aldrich), lithium hexafluorophosphate (Sigma Aldrich), and copper foil (MTI Corporation). The preparation of all solutions was carried with the help of distilled water. 

### 2.2. Activated Carbon Production

Lignin was extracted from sawdust of unclassified trees by treatment with a mixture of formic and acetic acids. A total of 100 mL of a mixture of formic and acetic acids (70:30) and 10 g of mixed sawdust were added to the flask. The mixture was stirred for 2.5 h at 115 °C in a magnetic stirrer. At the first stage, a rapid change in the color of the solution was observed. As the reaction progressed, the color became darker, and wood clusters also dissipated. The mixture was then cooled to 25 °C and filtered. The concentrated extraction solution was treated with water to precipitate lignin, in a ratio of 5:1, and the mixture was heated to 105 °C for 30 min. After adding water to the mixture, the color changed to light orange, and the formation of larger lignin agglomerates was easily seen, as the solution became more and more cloudy. 

The resulting lignin was extruded into lignin/PAN-based carbon fibers by an electrospinning unit. The obtained fibers were thermally stabilized in a tube furnace up to 220 °C, and during stabilization the fiber underwent cyclization, oxidation, and dehydrogenation. Carbonization of the stabilized fibers was then carried out.

### 2.3. Characterization

The morphology of the samples was found by scanning electron microscopy (SEM, JEOL, model JSM-6490LA). Solver spectrum instrument (NT-MDT) was used to measure Raman spectra with the help of the 473 nm laser. The laser beam was directed on the sample using a 100 × 0.75 NA Mitutoyo lenses, providing a laser spot < 2 μm in diameter. X-ray diffraction technique (XRD) with Cu Kα of λ = 1.54 Å radiation (Bruker) allowed for the characterization of the samples structurally. 

### 2.4. Electrochemical Measurements

#### 2.4.1. Preparation of Electrodes

The electrochemical performance of materials was investigated using a CR2032-type coin cell placed in an Ar-filled glove box (99.9999%). The working electrodes were prepared by evenly mixing CNF (800 °C, 900 °C, and 1000 °C), PVDF, and C45 in a weight ratio of 70:20:10. The uniform slurry was coated onto a pure Cu foil and dried at 110 °C for 12 h under vacuum. The working electrodes were punched into 13 mm diameter circular disks with an active-material loading of ∼1.2–2.3 mg cm^−2^. Pure lithium foil was used as the counter electrode and Celgard 2400 played the role of the separator. A 1M solution of LiPF_6_ in ethylene carbonate, dimethyl carbonate, and diethyl carbonate (EC + DMC + EMC, 1:1:1) was used as the electrolyte.

#### 2.4.2. Electrochemical Measurements

The galvanostatic charge/discharge measurements were carried using a BTS (CT-4008-5 V 10 mA-164) in the voltage range of 0.01−3 V (vs Li/Li+) at different mA. The cyclic voltammetry (CV) and electrochemical impedance spectroscopy (EIS) measurements were made using a Potentiostat Galvanostat P-40x electrochemical station. The EIS measurement was conducted in the frequency range of 0.1 Hz to 100 kHz at the open-circuit voltage. The cyclic voltammetry (CV) scan rate was 0.1 mV s^−1^ in the voltage range of 0.01−2.8 V.

## 3. Results and Discussion

### 3.1. Activated Carbon Surface Characterization

The morphology of the obtained carbon fibers was studied using SEM (Figure 1).

The fiber diameters vary from 60 nm to 170 μm (SEM images in Figure 1), and do not change substantially at different carbonization temperatures. An exception is LCF 800 °C, where more aligned fibers are observed (Figure 1a). This could be caused by different degrees of carbonization because of a lower temperature compared with fibers carbonized at 900 and 1000 °C (see Figure 1b,c). Depending on the constituents in the shell, this could influence the electrochemical performance. The XRD and Raman spectra of carbon fibers are shown in Figure 2 and Figure 3, respectively. 

To determine the composition of the crystalline and amorphous states of CNF, they were studied using powder X-ray diffraction patterns (Figure 2). The X-ray diffraction analysis of CNF illustrates the presence of two characteristic wide humps of carbon, including diffraction from {002} planes located at an angle of 24.7°, which indicates a disordered structure of carbon [34] and diffraction from planes {100} at an angle of 43.6° corresponding to the turbostratic carbon plane [35]. The broad shape of the peak around 24.1° indicates that the carbon nanofibers are mainly composed of amorphous carbon. The average distance between the layers {002} and {100} is calculated by the Bragg equation (Table 1).

From Figure 3, it is clearly seen that the degree of graphitization is calculated from the ratio of the peak area G to the total area of the spectrum in the range from 700 cm^−1^ to 2000 cm^−1^, and is ~22% in the case of 800 °C and 900 °C, and 26.7% in the case of annealing at 1000 °C.

Degree of graphitization is calculated by Formula (1)
(1)$$G_{f}=\frac{A\left(G\right)}{\sum_{500}^{2000}A}\cdot 100\%$$

G_f_—degree of graphitization;

A(G)—area of G peak;

∑A—full area of the spectrum.

### 3.2. Electrochemical Characterization 

Cyclic voltammetry of three different electrodes was recorded at a scan rate of 0.1 mVs^−1^ between 0.001 and 2.8 V, as shown in Figure 4a,c,e. The CV of CNF-based electrodes synthesized at 800, 900, and 1000 °C was tested after several charge–discharge cycles. Therefore, the samples do not show peaks in the formation of the SEI layer. All three cycles of the samples are almost the same. This indicates that the SEI layer has already been formed, and it protects the electrode well from direct contact with the electrolyte. During lithiation and delithiation, the SEI layer withstands the charging and discharging (reversible) reactions well. 

Galvanostatic charge/discharge profiles at room temperature were plotted from cycles 1 to 50 at a current density of 100 mA/g at a voltage of 0.01–3.0 V. All samples have a plateau in the first discharge cycle in the region of 0.7 V, and the capacity is much higher compared to the following cycles. The discharge capacity of the CNF800 at 100 mA/g shows 798 mAh/g and charge capacity 555 mAh/g (Figure 4b); for CNF900 at 100 mA/g, discharge capacity shows 668 mAh/g and charge capacity 354 mAh/g (Figure 4d); and for CNF1000 at 100 mA/g, shows around 594 mAh/g and the charge capacity of 303 mA/g (Figure 4f). Compared with other samples, CNF 1000 shows a slightly lower capacity. The difference in capacitances in the first cycle is due to the formation of an SEI layer on the surface of the electrodes. The coulombic efficiency for CNF1000 is 51%, CNF900 is 52%, and CNF800 is 69%. Starting from the second onto the next cycles, the capacity is stable and coulombic efficiency for all samples increases to almost ~98–99%. All three samples show good stable cycling at different current densities (Figure 4d).

However, the sample carbonized at 1000 °C shows a higher specific capacity of 594 mAh/g in comparison with the results, which demonstrate that the CNF anode obtained at 1000 °C exhibits a specific capacity up to 271.7 mAh/g [33].

To determine the kinetics of the assembled cells, electrochemical impedance spectroscopy (EIS) measurements were performed on half-cells based on CNF800, CNF900, and CNF1000 before cycling and after 100 cycles. Figure 5a shows the impedance of the fresh cells (before cycling). The Nyquist plot of the EIS curves with sinusoidal excitation signals is in a frequency range of 0.1 Hz to 100 kHz. Internal resistance (Rs) for 800 and 900 C is 2.4 Ohms, and for 1000 C is 3.6 Ohms. For all three samples, their own resistances are not very different (2.4–3.6 Ohm), indicating a high conductivity. The charge transfer resistance (Rct) for 800 C is 250 Ohms, for 900 C is 490 Ohms, and for 1000 C is 3.5 kOhm. This may be due to the fact that a passivating film has grown on the electrode surfaces and due to the structure of the samples. After 100 cycles, the internal resistance (Rs) of the three samples does not change much, while the charge transfer resistance (Rct) decreases greatly (Figure 5b). Compared with the first cycle, Rct becomes 19 Ohms for 800 °C, 40 Ohms for 900 °C, and 60 Ohms for 1000 °C. This may be due to the fact that the passivation film on the electrode surface is destroyed after cycling, and the materials of the electrode surface are restructured.

The Li+ diffusion coefficient of the CNFs is calculated by Formulas (2) and (3) [36]. The relationship between Z’ (real part resistance) and ω (angular frequency) is shown in Figure 6. 

The Warburg factor (σ) is determined from the slope, and is substituted using Equations (2) and (3) [36], where R is the gas constant; T is the absolute temperature; A is the surface area of the electrode; F is the Faraday constant; and C is the molar concentration of Li-ion in an active material.

With long cycles at a current density of 500 mA/g, the 1000 °C sample shows an increase in capacity after 110 cycles from 100 mAh/g, and stabilized at about 160–175 mAh/g at 200 cycles (Figure 7). However, after 400 cycles, the capacity begins to drop, and at 600 cycles it shows a capacity of around 120 mAh/g. The 900 °C sample shows a stable increase in capacity for the first 100 to 150 cycles, and consistently shows 180–200 mAh/g capacity up to 250 cycles. After that, it starts to drop and 600 cycles show 118 mAh/g of capacity. The 800 °C sample shows a drop from 420 mAh/g to about 350–370 mAh/g in the first 10 cycles, and 240 cycles show a capacity of 366 mAh/g after the capacitance begins to fall slowly. The Coulomb efficiency of all three samples remains around 100%. The regular drop in power (ovals) is related to the temperature regime of the room. During the day, the capacity drops due to the inclusion of the air conditioner and stabilizes at night (we turn off the air conditioner). For example, a decrease in the capacity of a sample of 800 °C is due to switching on an air conditioner between day and night.

To further demonstrate the difference in CNF obtained at different temperatures, SEM images and EDX analysis after 620 cycles for the electrodes are presented in Figure 8.

EDX analysis shows the content of carbon material in the sample at 800 °C as 31.44 wt%, 900 °C as 33.76 wt%, and 1000 °C as 54.47 wt%. The morphology of CNTs obtained at 800 °C (Figure 8a) is consistent with the performance in cyclic tests, and even after cycling, the electrode has a smooth surface. Figure 8b shows CNF obtained at 900 °C. Structural destruction (cracks) is observed on the surface, which is consistent with the performance in cyclic tests during long-term tests, the capacity of which show the smallest capacity compared to other samples. Figure 8c shows CNF obtained at 1000 °C. In comparison, the formation of cracks on the electrode is less than on the 900 °C electrode, but the appearance of dendrites on the surface of 1000 °C is observed.

## 4. Conclusions

The synthesized fibers based on sawdust were produced by an electrospinning method, and after carbonization at 800, 900, and 1000 °C, they were used to make electrodes for lithium-ion batteries. The temperature of carbonizations is shown to affect the capacity of the material. CNF carbonized at 800 °C shows a discharge capacity of 797 mAh/g, which is more than CNF carbonized at 900 °C, which shows a capacity of 668 mAh/g. Of the three samples, the smallest capacity is shown by the sample carbonized at 1000 °C, which shows 594 mAh/g. CNF-800 shows excellent performance and resistance to long cycles. This work has great prospects for obtaining inexpensive CNF obtained from sawdust as electrode materials for use in electrochemical storage devices.

## Figures and Tables

**Figure 1 nanomaterials-12-03710-f001:**
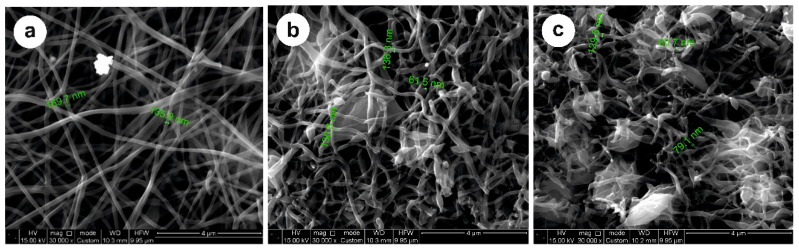
SEM images of carbon fibers at: (**a**) 800 °C; (**b**) 900 °C; (**c**) 1000 °C.

**Figure 2 nanomaterials-12-03710-f002:**
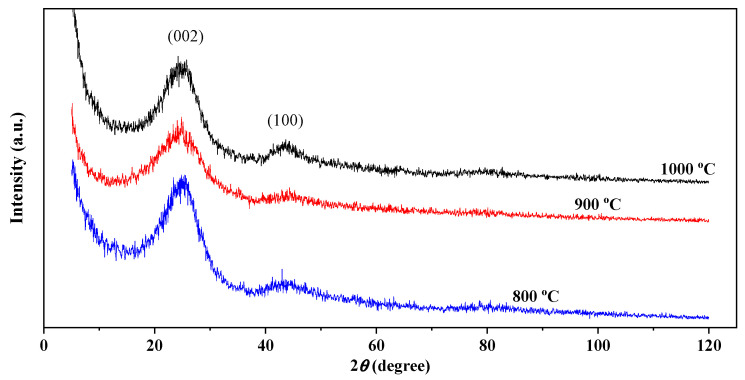
XRD of carbon fibers.

**Figure 3 nanomaterials-12-03710-f003:**
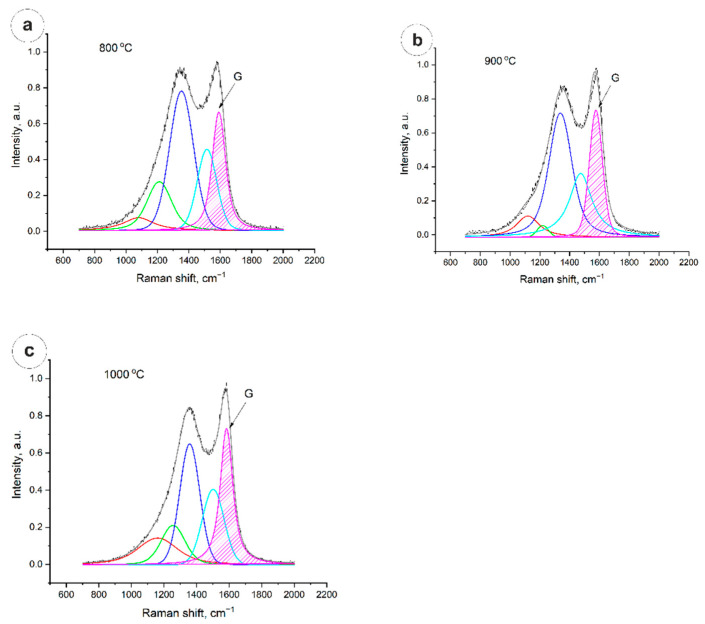
Raman spectra of carbon fibers: (**a**)—800 °C; (**b**)—900 °C; (**c**)—1000 °C.

**Figure 4 nanomaterials-12-03710-f004:**
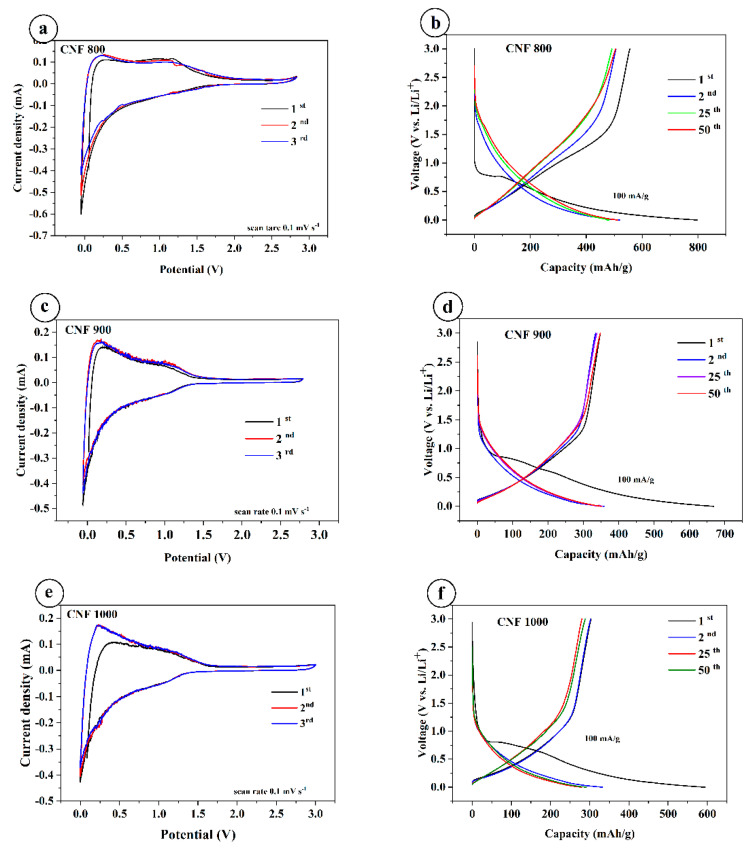
Cyclic voltammetry and charge–discharge curves of CNF electrodes: (**a**,**c**,**e**)—cyclic voltammetry curves; (**b**,**d**,**f**)—charge–discharge curves.

**Figure 5 nanomaterials-12-03710-f005:**
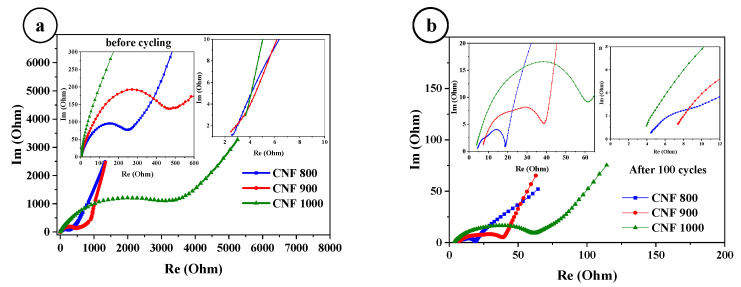
Impedance curves of CNF before, (**a**) and after cycling, (**b**).

**Figure 6 nanomaterials-12-03710-f006:**
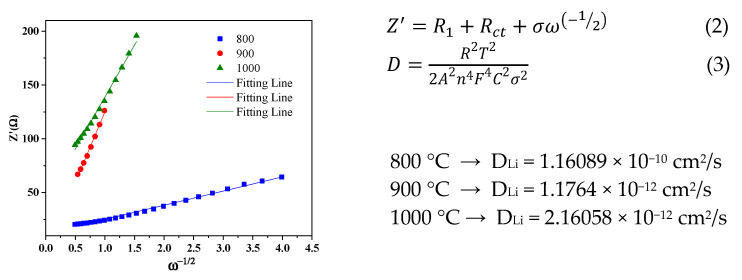
The relationship between Z’—real part resistance and ω—angular frequency.

**Figure 7 nanomaterials-12-03710-f007:**
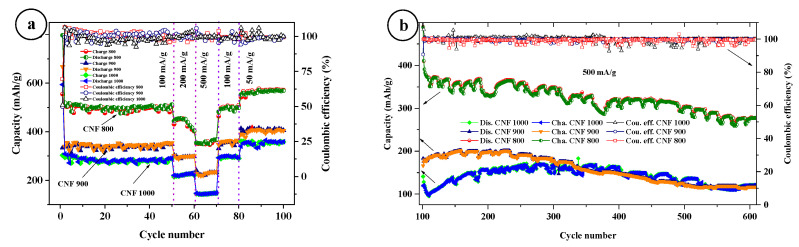
Cycling performance of CNFs at different currents (**a**), long cycles at a current density of 500 mA/g (**b**).

**Figure 8 nanomaterials-12-03710-f008:**
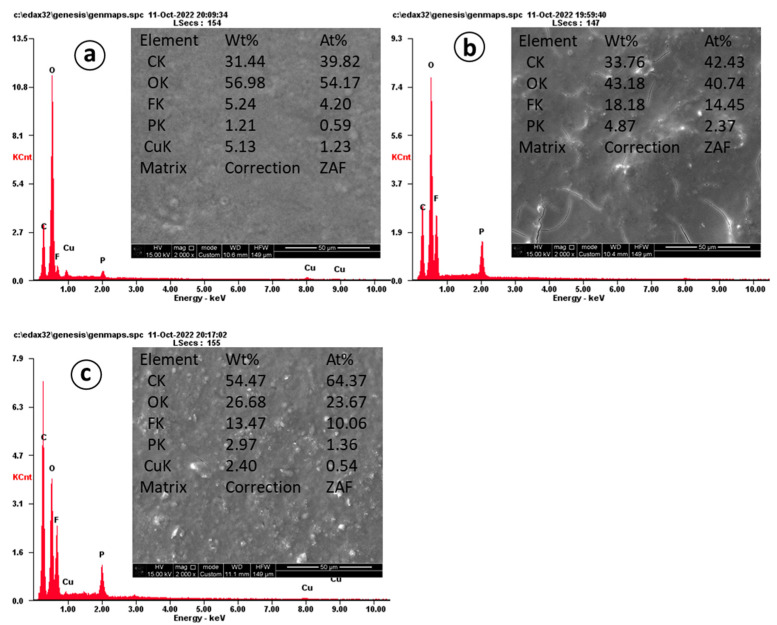
EDX and SEM images of CNF electrodes after 600 cycles: (**a**) 800 °C, (**b**) 900 °C, (**c**) 1000 °C.

**Table 1 nanomaterials-12-03710-t001:** Interlayer spacing parameters of the CNF.

Samples	{002}		{100}	
	2 θ	D Spacings, nm	2 θ	D Spacings, nm
CNF 800	24.7251	0.356245	43.6751	0.207083
CNF 900	24.9251	0.357655	43.8751	0.26185
CNF 1000	24.9751	0.35979	44.2251	0.204634

## Data Availability

Not applicable.

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
