# Peer review of "Biomass-Derived Porous Carbon Materials for Li-Ion Battery"

_nanomaterials, 2022, doi:10.3390/nano12203710_

Round 1
Reviewer 1 Report
1. Scale bar needs to identify clearly in images
2. Why does morphology change with temperature?
3. Slight shift in XRD peaks need to explain?
4. XPS analysis must provide to understand chemistry.
5. Figure quality is not acceptable?
6. Why does capacitance change with temperature?
7. EIS analysis must fit with physical quantity.
Author Response
- Scale bar needs to identify clearly in images
Author’s response:
We agree. All images in tif format as well as separately in additional files.
- Why does morphology change with temperature?
Author’s response:
After carbonization of carbon nanofibers at temperatures of 800°C, 900°C, and 1000°C, the morphology (structure) of the sample changes (destroys) with increasing temperature. At a temperature of 800°C, the fiber diameters are fairly uniform in the range of about 130-200 nm and the structure is retained. At 900 and 1000 C, due to the destruction of the structure, it is difficult to determine the diameters of the fibers. Changes in morphology may be the result of agglomeration due to greater particle mobility at higher temperatures.
https://doi.org/10.3390/fib5020022
DOI:10.1002/cssc.202000520
- Slight shift in XRD peaks need to explain?
Author’s response:
Most of the CNF is an amorphous carbon material that is confirmed by X-ray diffraction. As the temperature rises, the graphitization of the structure of the carbon nanofiber occurs. The Raman spectrum also confirms this. Graphitization of samples increases with temperature increase from ~ 22% to 26.7%. It is known that the diffraction angle (2θ) of the 002 peak of crystalline graphite is 26.56 â—¦ but the 002 peak in this figure was located at about 24.7 â—¦ at 800 C and slightly increased to 24.97 â—¦ at 1000 C with CNF graphitization.
DOI:10.1590/S1517-70762007000300009
10.1557/JMR.2010.0207
https://doi.org/10.1016/j.cartre.2021.100071
- XPS analysis must provide to understand chemistry.
Author’s response:
XPS analysis would show the exact relationship, but unfortunately we do not have such equipment.
- Figure quality is not acceptable? The Figures should be refined and well organized.
Author’s response:
All images in tif format as well as separately in additional files.
- Why does capacitance change with temperature?
Author’s response:
The nanofibers obtained at 800 C showed the highest capacity compared to the other 900 and 1000 C. As mentioned above, the sample obtained at 800 C carbonization, the SEM images clearly show that the structure of the nanofibers is solid. However, when the temperature rises at 900 and 1000 C, the structure of nanofibers is destroyed and agglomeration of nanofibers appears. Most likely, this affected the capacity of the samples, that is, when the structure of the samples is destroyed, the conductivity of the material worsens, and this has an effect on the charge transfer. In impedance measurements, the 900 and 1000 C samples showed high Rct.
- EIS analysis must fit with physical quantity.
Author’s response:
Nyquist plot of the EIS curves with sinusoidal excitation signals is in a frequency range of 0.1 Hz to 100 kHz. Internal resistance (Rs) for 800 and 900 C is 2.4 Ohms for 1000 C 3.6 Ohms. For all three samples, their own resistances are not very different (2.4-3.6 ohms), this indicating a high conductivity. Charge transfer resistance (Rct) for 800 C is 250 Ohms, for 900 C is 490 Ohms and for 1000 C is 3.5 kOhm. This may be due to the fact that a passivating film has been grown on the electrode surfaces and due structure of samples. After 100 cycles, the Internal resistance (Rs) of the three samples did not change much, the charge transfer resistance (Rct) decreased much (Figure 5 b). Compared with the first cycle, Rct became 19 ohms for 800 °C, 40 ohms for 900 °C and 60 ohms for 1000 °C. This may be due to the fact that the passivation film on the electrode surface is destroyed after cycling and the materials of the electrode surface are restructured.

Reviewer 2 Report
Recommendation: major revision.
Comments: In this work, the authors a biomass-based carbon nanofibers (CNF) were synthesized using lignin extracted from sawdust and polyacrylonitrile (PAN) via an electrospinning method and followed by a pyrolysis process. The structure and physicochemical properties of as-prepared materials were studied in detail with suitable techniques and reasonably explained. The electrochemical performance of this CNF anode for LIBs is investigated. The experiment data relevant to biomass-based anode offered in this manuscript are sufficient to support the conclusion. So, I recommend that this manuscript may be accepted for publication in Nanomaterials after major revision.
1. The introduction of this paper needs to make a strong argument about the impact and novelty of the work further. So, the introduction should enrich some lithium anodes in this section and demonstrate the advantages of this CNF anode over the commercial graphite anode.
2. The Figures should be refined and well organized.
3. To evaluate these biomass-based CNF anodes undergo, the SEM or TEM images after cycles are better offered.
4. The cycling performance and corresponding coulombic efficiency seem not stable in Figure 6. the authors should explain it.
5. The Li+ diffusion coefficient of this biomass-based CNF anode is better calculated.
6. The authors better compare the electrochemical performance of the biomass-based CNF anode with reported biomass-anode materials.
7. Some issues and writing mistakes exist in the manuscript. The authors should carefully check and correct them. Such as “scan rate was 0.1mV” in the 145th Line should be“scan rate was 0.1mV s-1 ”, “LiPF6 in ethylene carbonate” in the 137th Line should be “LiPF6 in ethylene carbonate”

Author Response
- The introduction of this paper needs to make a strong argument about the impact and novelty of the work further. So, the introduction should enrich some lithium anodes in this section and demonstrate the advantages of this CNF anode over the commercial graphite anode.
Author’s response:
We agree. We have added to the manuscript the paragraph below:
Graphite is currently the most successfully commercialized anode material. However, its limited theoretical capacity and limited power density seem to be insufficient for the next generation LIB. To overcome these problems, new materials with fundamentally higher capacitance and higher power density are urgently needed. Recently, there has been a growing interest in the development of new carbon nanomaterials to replace graphite as anode materials for LIB. These materials include carbon nanofibers obtained by us from wood waste and PAN polymer by electrospinning.
- The Figures should be refined and well organized.
Author’s response:
All images in tif. format as well as separately in additional files.
- To evaluate these biomass-based CNF anodesundergo, the SEM or TEM images after cycles are better offered.
Author’s response:
Unfortunately, we do not have TEM equipment. But, we have possibility to investigate fibers by SEM only before we made electrodes.
- The cycling performance and corresponding coulombic efficiency seem not stable in Figure 6. the authors should explain it.
Author’s response:
We have checked and fixed, these are most likely technical errors when transferring the database from tester to origin.
- The Li+diffusion coefficient of this biomass-based CNF anode is better calculated.
Author’s response:
We agree.
- The authors better compare the electrochemical performance of the biomass-based CNF anodewith reported biomass-anode materials.
Author’s response:
We agree. After Fig.5. we have added to the manuscript the paragraph below:
However, carbonized at 1000 °C sample showed higher specific capacity 594 mAh /g in comparison with the results demonstrated that the CNF anode obtained at 1000 °C exhibits a specific capacity up to 271.7 mAh /g [33].
- Some issues and writing mistakes exist in the manuscript. The authors should carefully check and correct them. Such as “scan rate was 0.1mV” in the 145th Line should be“scan rate was 0.1mV s-1”, “LiPF6 in ethylene carbonate” in the 137th Line should be “LiPF6 in ethylene carbonate”
Author’s response:
Thanks for showing our mistakes, we will fix all the mistake

Round 2
Reviewer 2 Report
Recommendation: major revision.
In this work, the authors report a biomass-based carbon nanofiber (CNF) was synthesized from sawdust and polyacrylonitrile (PAN) via an electrospinning method followed by a pyrolysis process as an anode for lithium-ion batteries. The structure and physicochemical properties of as-prepared CNFs are offered, but there lack mechanism investigation and structure evolution in the manuscript. Though the authors addressed the questions point to point according to previous reviewers, there exist some issues that need further to be solved. To meet the high criterion of Nanomaterials, this manuscript must be a major revision.
1. Figure 3 is still not clear enough.
2. To evaluate these biomass-based carbon nanofibers (CNF) undergo, the SEM images after cycles are better offered.
3. The Li+ diffusion coefficient of the biomass-based carbon nanofibers (CNF) is better calculated.
4. The excellent electrochemical performance should be reflected in the full battery, so if the author better added the full battery data.
Author Response
Recommendation: major revision.
In this work, the authors report a biomass-based carbon nanofiber (CNF) was synthesized from sawdust and polyacrylonitrile (PAN) via an electrospinning method followed by a pyrolysis process as an anode for lithium-ion batteries. The structure and physicochemical properties of as-prepared CNFs are offered, but there lack mechanism investigation and structure evolution in the manuscript. Though the authors addressed the questions point to point according to previous reviewers, there exist some issues that need further to be solved. To meet the high criterion of Nanomaterials, this manuscript must be a major revision.
- Figure3 is still not clear enough.
Author’s response:
We have changed Fig.3. as we could.
- To evaluate these biomass-based carbon nanofibers (CNF) undergo, the SEM images after cycles are better offered.
Author’s response:
SEM images after cycles was provided in manuscript as Fig.8.
Figure 8. EDX and SEM images of CNF electrodes after 600 cycles a) 800 °C b) 900 °C c) 1000 °C
EDX analysis showed the content of carbon material in the sample at 800 °C 31.44 wt%, 900 °C 33.76 wt%, 1000 °C 54.47 wt%. The morphology of CNTs obtained at 800 °C (Figure 8 a) is consistent with the performance in cyclic tests, even after cycling the electrode has a smooth surface. Figure 8 b) shows CNF obtained at 900 °C. Structural destruction (cracks) is observed on the surface, which is consistent with the performance in cyclic tests during long-term tests, the capacity of which showed the smallest capacity compared to other samples. Figure 8 c) shows CNF obtained at 1000 °C. In comparison, the formation of cracks on the electrode was less than on the 900 °C electrode, but the appearance of dendrites on the surface of 1000 °C is observed.
- The Li+diffusion coefficient of the biomass-based carbon nanofibers (CNF) is better calculated.
Author’s response:
The Li+ diffusion coefficient of the CNFs was calculated by formula below and added to the text of manuscript.
Also the relationship between Z’ - real part resistance and ω - angular frequency was drawn and included to the manuscript as Fig.6.
|
800 °Ð¡ → DLi=1,16089*10-10 cm2/s 900 °Ð¡ → DLi=1,1764*10-12 cm2/s 1000 °Ð¡ → DLi=2,16058*10-12 cm2/s |
Figure 6. The relationship between Z’ - real part resistance and ω - angular frequency
- The excellent electrochemical performance should be reflected in the full battery, so if the author better added the full battery data.
Author’s response:
To obtain a complete analysis of the batteries, we will need to assemble a full cell, but unfortunately, at the moment our institute is closed for cumulative repairs and therefore we will not be able to assemble a full battery and electrochemical measurements at the moment.

Round 3
Reviewer 2 Report
The quality of the manuscript has been improved. It is now publishable.